# Toward Understanding the Mechanisms of Malignant Peripheral Nerve Sheath Tumor Development

**DOI:** 10.3390/ijms22168620

**Published:** 2021-08-10

**Authors:** Teddy Mohamad, Camille Plante, Jean-Philippe Brosseau

**Affiliations:** 1Department of Biochemistry and Functional Genomics, Faculty of Medicine and Health Sciences, Université de Sherbrooke, Sherbrooke, QC J1E 4K8, Canada; Teddy.Mohamad@usherbrooke.ca (T.M.); Camille.Plante@usherbrooke.ca (C.P.); 2Centre de Recherche du Centre Hospitalier Universitaire de Sherbrooke, Université de Sherbrooke, Sherbrooke, QC J1E 4K8, Canada

**Keywords:** neurofibromatosis type I, malignant peripheral nerve sheath tumor, neurofibroma, mouse models, tumor microenvironment

## Abstract

Malignant peripheral nerve sheath tumors (MPNSTs) originate from the neural crest lineage and are associated with the neurofibromatosis type I syndrome. MPNST is an unmet clinical need. In this review article, we summarize the knowledge and discuss research perspectives related to (1) the natural history of MPNST development; (2) the mouse models recapitulating the progression from precursor lesions to MPNST; (3) the role of the tumor microenvironment in MPNST development, and (4) the signaling pathways linked to MPNST development.

## 1. Natural History

By definition, malignant peripheral nerve sheath tumors (MPNSTs) are sarcomas originating/arising from a peripheral nerve. This broad definition encompasses 5–10% of all sarcomas. One of the main risk factors is having a neurofibromatosis type I (NF1) diagnosis [1]. Still, around 45% of MPNST cases arise sporadically, and 10% are due to previous radiation exposure. Most MPNST patients are diagnosed in their thirties with no gender biases. Because MPNSTs are tumors affecting sensitive nerves, it is not rare for these patients to experience numbness in their extremities or having difficulty moving their limbs (paraparesis).

Plexiform neurofibroma (pNF) is a well-characterized benign precursor lesion that is usually closely followed up in NF1 patients for any sign of transformation to an MPNST (as opposed to a sporadic MPNST, where a pNF is not suspected *a priori*). The bulk of pNF growth occurs mainly in childhood, so any significant growth in adulthood should raise concerns about malignant progression. Magnetic resonance imaging can assist in capturing the dynamic evolution of MPNST. Surgical removal remains the standard of care. Adjuvant radiotherapy or chemotherapy is also widely administered, although the added benefit is a matter of debate [2,3]. Even so, the prognosis for MPNST-bearing NF1 patients is worse than that for sporadic MPNST patients. This may be proportional to the difference in tumor size and body location distribution, as MPNST from NF1 patients tends to be larger and deeply located, and hence challenging to resect with clear margins.

The genomic architecture of MPNST is similar to other soft tissue sarcomas, with an elevated frequency of copy-number alterations and a relatively modest burden of single nucleotide variants [4]. Aside from *NF1* microdeletion, a large deletion of 1.4 Mb encompassing the *NF1* gene [5,6,7] occurring in up to 10% of NF1 patients [8], there is no specific, highly recurrent chromosomal abnormality in MPNST cancer cells. Interestingly, patients with *NF1* microdeletion have a 16–26% risk of developing MPNST [9] (compared to an 8–13% risk of developing MPNST for NF1 patients regardless of the specific *NF1* mutation [10]). There are at least 14 protein coding genes and four miRNAs encoded in the microdeletion region. This led to the hypothesis that the increased MPNST risk may be due to the loss of additional tumor suppressor genes. A decade of effort culminated with the demonstration that SUZ12 loss, one of the 14 protein coding genes, promotes MPNST progression in mouse models [11]. In fact, mutations in components of the polycomb repressive complex 2 (PRC2) complex (i.e., EED and SUZ12) are associated with MPNST [12,13]. The majority of MPNSTs also harbor somatic mutations in the cell cycle regulator *CDKN2A*, regardless of their etiology [4]. Current evidence indicates that mutation in these genes follows a specific order (Figure 1). Inactivation of the *NF1* gene in the Schwann cell lineage promotes neurofibroma development in a nerve plexus. The *NF1*^−/−^ benign neurofibroma Schwann cells are relatively slow-proliferating and do not invade adjacent tissue structures, although they can be large and weigh several kilograms. Inactivating mutation in the cell cycle regulator locus *CDKN2A* subsequently promotes a neurofibroma’s progression into a pre-malignant state called atypical neurofibroma of uncertain potential (ANNUBP). Further mutations in the PRC2 complex lead to an MPNST. The PRC2 complex is responsible for histone 3 lysine 27 (H3K27Me3) methylation and is critical to normal development. In Schwann cells, H3K27Me3 represses key nerve repair genes, and inactivation of the PRC2 component EED activates them [14]. Moreover, EED loss causes impaired Schwann cell proliferation after injury, with significant induction of the *Cdkn2a* [14]. Thus, it is tempting to speculate that the loss of PRC2 function in MPNST leads to the activation of key nerve repair genes but does not impair proliferation due to mutation in the CDKN2A locus. The following most frequently mutated gene is *TP53*. However, the *TP53* mutation rate is relatively low and does not necessarily undergo loss-of-heterozygosity in MPNST [15]. It is unclear whether *TP53* mutation is a late event or part of a distinct tumorigenic path. Recurrent mutations in *TSC1/2* were recently reported, but these results have not been confirmed by any other group so far [4]. *PTEN* is not frequently mutated, but its expression was lower in MPNST than in neurofibroma in two independent cohorts [16,17]. The contribution and/or therapeutic relevance of infrequent mutations in Ras-associated pathways [18] also remains to be established.

Kehrer-Sawatzki et al. [8] pointed out that there are at least three other genes with tumor suppressor activity within the microdeletion region: miR193A, miR365B, and the ATPase family AAA domain-containing protein 5 (ATAD5). Although there is a wide body of evidence of their tumor suppressor activity in other cancer types, there is currently no report of a direct contribution of any of these three genes in the context of MPNST development. Importantly, there could be a significant position effect on genes flanking the microdeletion that may ultimately promote MPNST progression. For example, a microdeletion produces a chimeric gene involving SUZ12 that results in SUZ12 and surrounding genes being overexpressed [19].

MPNST is particularly prone to metastasis [20,21]. Head and neck metastasis is associated with a worse prognosis, whereas non-head and neck, such as lung metastasis, is associated with better survival [21]. We still do not know if there are additional mutations/chromosomal changes in metastatic vs. primary MPNST tumors. Chemotherapy (doxorubicin) is the standard of care for metastatic MPNST [22].

In summary, several questions remain unanswered, and some of them may ultimately converge on the exact identity of the cell of origin of MPNSTs. It is assumed that most MPNSTs originate from the Schwann cell lineage, although some may have a fibroblastic origin [23,24,25]. The origin of epithelioid MPNST is unclear. It is also unclear if all MPNSTs progress from pNF/ANNUBP or if some can appear *de novo*. In the latter case, what would be the underlying genetic path?

## 2. Mouse Models

The most widely used mouse model recapitulating MPNST is the *cisNf1*^+/−^*p53*^+/−^ model [26,27], which represents a milestone in the NF1 field. However, it suffers from several drawbacks such as its low penetrance for MPNST (30%), its frequent manifestation as a subcutaneous tumor without macroscopic nerve connection or an apparent precursor lesion (i.e., neurofibroma), and not being genetically representative of the majority of MPNSTs in the clinic. To engineer a model that mimics the pNF (benign) to MPNST (malignant) sequence, Hirbe et al. decided to build on the *Postn*-cre *Nf1*^f/f^ and *Gfap*-cre *Nf1*^f/f^ neurofibroma models by adding an inducible system to knockdown p53 [28]. Upon neurofibroma formation (60–70% penetrance), subsequent repression of p53 catalyzes the development of MPNST in a stepwise manner [28]. This approach could be used to characterize other relevant tumor suppressor genes frequently mutated in human MPNST (e.g., CDNK2A, PRC2 component) after pNF establishment in mice.

Several xenograft models where human MPNST cell lines are implanted subcutaneously or in the vicinity of the peripheral nervous system models have been reported [29,30,31]. However, there is no report of human cells representing precursor lesions (i.e., neurofibroma) implanted into the peripheral nervous system that spontaneously leads to MPNST development. The only reported mouse model so far that forms neurofibroma upon *Nf1* inactivation and spontaneously leads to MPNST, later on, is the *Plp*-creERT2; *Nf1*^f/f^ model [32]. In this case, MPNSTs were characterized by the loss of H3K27Me3 and S100 markers. However, it is not clear whether the actual genetic lesion (if any) that occurs somatically drives the malignant progression in this particular model.

Based on the notion that the Schwann cell growth factor neuregulin-1 and its receptor EGFR are expressed in neurofibroma and MPNST [33,34,35], mice models overexpressing neuregulin-1 [36] or EGFR [37] were developed. Strikingly, overexpression of neuregulin-1 by itself (*Nf1* wild type) is sufficient to induce neurofibroma formation followed by MPNST development, although at low frequency [38]. Again, it is unclear whether additional genetic lesions (if any) occur somatically and drive the malignant progression in this particular model. Overexpressing EGFR in *Nf1* wild-type Schwann cells leads to neurofibroma formation, although to a low penetrance, but not to MPNST [37]. The *Dhh*-cre *Nf1*^f/f^ mice form neurofibromas with 100% penetrance but this does not lead to spontaneous MPNST formation [39]. Interestingly, this model does so when coupled to EGFR overexpression (i.e., *Dhh*-cre; *Nf1*^f/f^; CNPase-*EGFR*) [40] or *Pten* deletion (i.e., *Dhh*-cre; *Nf1*^f/f^; *Pten*^f/f^; [41]). *Gfap*-cre *Pten*
^f/+^ LSL KrasG12D mice develop MPNST with 100% penetrance. Evidence indicates that *Pten* undergoes loss-of- heterozygosity in the latter model, but *Nf1* and *p53* remain wild type.

Until recently, no models could recapitulate ANNUBP lesions, the closest precursor lesion to MPNST (Figure 1). To develop such a model, two independent groups coupled their neurofibroma model to the inactivation of the *Arf* gene (*Cdkn2a* locus) [42,43]. The *Dhh*-cre *Nf1*^f/f^
*Arf*
^+/−^ mice develop ANNUBP lesions in precursor neurofibroma tumors but remain benign [42], whereas the ANNUBP lesions of the *Postn*-cre *Nf1*^f/f^
*Arf*
^f/f^ further develop into MPNST with high penetrance [43]. Interestingly, injection of adenovirus-Cre in the sciatic nerves of *Nf*
^f/f^
*Arf*
^f/f^ leads to MPNST development without an apparent precursor lesion [44].

As summarized in Table 1, there is no consensus on the histopathological characterization of mouse models of MPNSTs. According to a recent NF expert consensus, human MPNST histological characteristics include the following: (1) confirmation of a neoplasm of Schwann cell (presence of S100+ cells or S100+ micronerve); (2) hypercellularity, (3) cytologic atypia; (4) loss of neurofibroma architecture; (5) a mitotic index of more than 10/10 HPF or more than 3/10 HPF with a necrotic area [45]. However, these criteria developed for human MPNST were not systematically applied to characterize mouse MPNST (Table 1). Most research groups perform immunohistochemistry with anti-S100 antibodies to demonstrate that there are at least a few cells positive for the Schwann cell marker S100 (as supporting evidence for a Schwann cell origin) and acknowledge hypercellularity. However, these few features by themselves are not sufficient to discriminate MPNST from ANNUBP [45]. Consequently, it is not easy to compare between models and to distinguish between ANNUBP and MPNST. In fact, there is a need to develop a set of immunohistochemical markers discriminating the progression of the neurofibroma to MPNST sequence in both mouse models and human samples. Recently, the loss of H3K27Me3 was validated as an MPNST marker by several independent groups in mouse and human MPNST [32,46,47,48]. Interestingly, there is a concomitant gain of acetylation at the H3K27 position in MPNST cancer cells, making them sensitive to protein binding acetylated histone [11]. However, it is unclear if H3K27Ac can be used as a positive IHC marker to discriminate an MPNST from a neurofibroma. That would be helpful, as the current set of MPNST histological markers is mostly limited to relatively subjective features observed on hematoxylin and eosin staining (cytologic atypia, mitotic figures, hypercellularity, overall architecture) [45]. Adjacent tissue invasiveness [26,28,42] and Ki67 positivity [28,42,44] were also used to discriminate neurofibroma over MPNST. Other markers such as the loss of CD34 lattice [43] and the presence of mast cells [44] were used to complement the histological analysis of the neurofibroma to MPNST sequence but warrant further study.

## 3. Tumor Microenvironment

From a developmental point of view, the peripheral nervous system originates from the neural crest. Upon neural tube closing, some neural crest cells migrate laterally (becoming melanocytes), whereas others migrate to initially form the dorsal root ganglion and eventually a nerve by migrating through immature connective tissues [49]. A mature nerve fiber comprises several bundles of axonal projection co-existing with fibroblasts, endothelial cells, and immune cells. Physiologically, the Schwann cells wrap the axons of the peripheral nerves and secrete the myelin sheath. In neurofibroma, the same cell types co-exist but in a disorganized manner [50,51]. Given this context, one may think that the origin of the neurofibroma cells composing the microenvironment comes from the nerve, but it is unclear. Neural crest stem cells can generate nerve fibroblasts in addition to Schwann cells in normal tissue [52], but the origin of neurofibroma fibroblasts may be different. Because loss-of-heterozygosity does not happen in neurofibroma fibroblasts [53,54,55], it leaves at least two possibilities: *NF1* loss-of-heterozygosity (1) prevents the accumulation of neural-crest stem cell-derived fibroblast or; (2) occurs in a glial-restricted progenitor. Of note, neurofibroma developing in the skin dermis (cNF) has identical histology compared to neurofibroma developing in nerve plexus (pNF), suggesting that they may have a common neural origin. These two neurofibroma types have different clinical presentations. cNF begins to manifest at puberty and has zero malignant potential, whereas pNF can be congenital and frequently progress to MPNST. Perhaps this could be explained by the influence of skin dermis cells and/or a different cell of origin.

Cataloging the cell of origin of each microenvironment cell from neurofibroma is challenging because there is no clear marker discriminating each cell type. Recently, Brosseau et al. began to fill this gap by performing single-cell RNAseq (scRNAseq) on human neurofibroma samples [50]. They discovered a shortlist of potential cell type-specific markers. It remains to be established if these markers can also distinguish neurofibroma microenvironment cells from normal cells and MPNST microenvironment.

The variation in the extracellular matrix and cellular composition of the tumor microenvironment as it progresses in the neurofibroma to MPNST sequence has been little explored. S100 marks Schwann cells in neurofibroma, but this marker is often lost in MPNST. This is most probably due to a decreased expression of Schwann cell differentiation marker along with cancer progression. Early on, CD34+ S100-cells were identified in the microenvironment of neurofibroma [56]. Later, CD34 was proposed as a neurofibroma fibroblast marker [57], although CD34+ cells are found in normal tissues [56]. As neurofibroma progresses to ANNUBP and MPNST, CD34 signal heterogeneity increases. There is a trend toward a lower and more discrete CD34 staining by immunohistochemistry, but the small sample size has precluded definitive conclusions [45,57,58]. Whether CD34+ fibroblasts simply decrease their CD34 expression or change population size is unclear. Besides CD34, neuron-glial antigen 2 (NG2) was also reported as a nerve fibroblast marker but was not further validated in neurofibroma or MPNST [59]. More than a dozen potential neurofibroma fibroblasts distinguishing from normal skin fibroblasts were discovered recently, but their specificity in relation to normal nerve and MPNST was not investigated [50]. Surprisingly, this list does not include the classic fibroblast marker alpha smooth muscle actin (αSMA) [50], suggesting that neurofibroma fibroblasts are distinct from cancer-associated fibroblasts and myofibroblasts found in major organ fibrosis. No other fibroblast marker was reported for MPNST. Hopefully, applying scRNAseq technology to ANNUBP and MPNST samples will yield insights into novel MPNST fibroblast markers.

Although neurofibroma fibroblasts are negative for αSMA, it is expressed by neurofibroma pericytes, an important cell type that wraps endothelial cells. Using αSMA to stain the microvasculature of neurofibromas and MPNSTs, Gesundheit et al. discovered that the meshwork structure is much more irregular as the tumors progress [60]. Targeting the pericyte marker NG2 reduced angiogenesis in a xenograft model of MPNST [61]. Historically, there was an initial interest in studying the contribution of pericytes and endothelial cells in neurofibroma and MPNST, but disappointing results using anti-angiogenic molecules in clinical trials [62,63] discouraged NF1 researchers from pursuing this path further. This picture may change in light of the recent success of cabozantinib, which may be attributed at least in part to its anti-angiogenic properties [64].

Neurofibromas and MPNSTs are composed of various immune cells. Mast cells are histamine-rich cells present in the connective tissue that play a key role in an allergic response. NF1 inactivation in Schwann cells leads to an increase in the stem cell factor, a mast cell-activating factor [65]. The presence of mast cells is a neurofibroma hallmark and can be identified by staining for the stem cell factor receptor (KIT). However, drastically reducing the number of mast cells does not impair mouse neurofibroma development [66], and therapeutic strategies targeting mast cells in MPNST patients largely failed [67]. A trend toward a lower mast cell count in MPNST [68,69] has been challenged [44]. Thus, the functional role of mast cells in the neurofibroma to MPNST sequence may be marginal.

Macrophages are infiltrating white blood cells with phagocytic capacities. They are found in high density in mouse and human neurofibromas and even more in MPNSTs [66,70]. Transcriptomic analysis performed on CD11b+ F4/80+ CD31- FACS-sorted cells (i.e., macrophages) revealed that neurofibroma macrophages are a mixed population of M1 (pro-inflammatory) and M2 (anti-inflammatory) [71]. Initial investigation suggests that M2 macrophages are less present in MPNST, but more work is needed to confirm this result [70].

T cells, especially CD8+ T cells, become activated upon binding to their receptors and deploy multiple mechanisms against tumor cells. Interestingly, *Nf1*^+/−^ CD8+ T cells are hyperactive compared to their wild-type counterpart [32]. This suggests that NF1 patients (as opposed to sporadic MPNST) would have an advantage in fighting against cancer, but this idea does not translate in the clinic, meaning that other mechanisms may counterbalance. One such mechanism is the expression of the immune checkpoint. Even if there is considerable heterogeneity in the level of CD8+ T cells and an incomplete understanding of the immune checkpoint targets expressed in neurofibroma and MPNST [72,73], clinical trials are actively running (e.g., EUCTR-2015-004747-39, NCT02691026) and show promising results [74]. There is a need to identify the most susceptible NF1 patients (i.e., hot tumor) and match them to the correct immune checkpoint blockers. Thus, inhibition of the critical immune checkpoint blockers to restore CD8+ T cell cytotoxic activity should be sufficient to negatively impact MPNST development, particularly in NF1 patients (over patients with sporadic MPNST).

As discussed by Brosseau and collaborators [75,76], it is known that the *Nf1*^+/−^ microenvironment contributes to neurofibroma development, but the underlying molecular and cellular mechanisms are elusive. Even less is known about the contribution of the *Nf1*^+/−^ microenvironment on MPNST development. Initial studies indicate that an *Nf1*^+/−^ microenvironment promotes MPNST development [44,77]. A CD45+ population of immune cells may possess tumor-promoting capacities [44]. However, a recent study using two orthogonal mouse models indicates that the *Nf1*^+/−^ microenvironment impairs malignant progression [32]. Because *Nf1*^+/−^ CD8+ T cells trigger an enhanced immune reaction and *Nf1*^+/−^ natural killer cells have a protective effect compared to their wild type counterpart, it raises the intriguing possibility that *Nf1*^+/−^ T cells may explain the impairing phenomenon observed by Brosseau et al. [32]. More works are needed to dissect the anti- and pro-tumorigenic properties of the *Nf1*^+/−^ microenvironment with respect to MPNST development.

## 4. Signaling Pathways

Cellular homeostasis is the ability to maintain a relatively stable and dynamic equilibrium of physical and chemical parameters allowing cellular process development and cellular integrity. Identifying the critical pathways for the progression from pNF to MPNST is essential to understanding malignant transformation and advanced therapy.

RAS proteins are core regulators of cellular signaling because of their crossroad position with many signal transduction pathways and are highly conserved across species [78,79]. RAS conformation oscillates between a GTP-coupled activated form and a GDP-associated inactivated form tightly controlled by GTPase-activating proteins (GAP). RAS signal transducers, once activated by tyrosine kinase receptors in response to growth factors, can both activate MAPK and mTOR pathways and promote cell development and proliferation (Figure 2). Although there is no oncogenic mutation in the Ras proteins of MPNST patients, the three Ras proteins (H-Ras, N-Ras, and K-Ras) are activated in MPNST cell lines [80]. Instead, NF1 negatively regulates RAS through its GAP activity, and its loss-of-function mutations lead to constitutive activation of RAS.

### 4.1. MAPK

One of the pathways controlled by RAS is the MAPK signaling pathway. This evolutionarily conserved pathway amongst eukaryotes regulates many cellular processes, including proliferation and cell cycle entry signals. Following growth factor stimulation, GTP-bound RAS activates a cascade of phosphorylation leading to RAF, MEK1/2, and ERK1/2 phosphorylation. Most MPNSTs do not directly harbor mutations in the MAPK pathway except for the occasional BRAF V600E oncogenic mutation (see references in [81]). In MPNST cell lines, the MAPK pathway is activated as judged by a Ras pull-down assay and Western-blot for b-Raf and c-Raf [82]. In a large retrospective analysis of an MPNST patient cohort (*n* = 140), more than 90% of the tumors were positive for *p*-MEK [83]. Similar results were obtained from an independent group [84]. Of note, MEK inhibition is effective by itself on pNF [85] and shows promising results in MPNST [86]).

### 4.2. PI3K/Akt/mTOR

Downstream of Ras, the mTOR pathway is also at the intersection of many pathways. mTOR and mLST8 form the mTORC1 and mTORC2 complexes when associated with raptor and rictor, respectively. mTOR signaling is involved in controlling cell growth, survival, proliferation, cytoskeletal organization, and autophagy [87]. In response to growth factor receptor activation, the phosphatidylinositol 4,5 biphosphate (PIP2) is phosphorylated and converted to phosphatidylinositol 3–5 triphosphate (PIP3) by the phosphatidylinositol 3 kinase (PI3K). The PIP3 leads to the phosphoinositide-dependent kinase 1 (PDK1) activation, and PDK1 proceeds to the Akt activating phosphorylation. Following the Akt activation, TSC1/2 and TBC1D7 complex, which constitutes the main negative regulator of mTORC1 activity, is inhibited. This double inhibition ultimately activates mTORC1. PI3K also can activate mTORC2. Once activated, mTORC2 can interact with Akt, phosphorylate it and consolidate the mTORC1 activation. Without stimulation, rictor is phosphorylated by glycogen synthase kinase 3, leading to rictor polyubiquitination and then degradation by the proteasome. Activation of Akt mediated by the PI3K strengthens rictor stability and thus mTORC2 activity. To turn off the mTOR signaling pathway, the phosphatase and tensin homolog (PTEN) decreases the PIP3 level, converting PIP3 to PIP2. Indeed, PTEN is involved in chromosomal stability and can promote cell cycle arrest by interacting with TP53.

About 50% of a large cohort of MPNST samples were positive for the phosphorylated form of Akt and mTOR (*p*-Akt and *p*-mTOR) [84]. Moreover, the prognostic analysis revealed that *p*-AKT, and *p*-mTOR, expressions were associated with poor prognosis [84]. Targeting mTORC1 (everolimus) reduced MPNST growth by 19–60% in a xenograft model [88] but was largely ineffective in clinical trials [63]. Because of the known MAPK and mTOR pathway cross-talk, a clinical trial (NCT03433183) aimed at determining the efficacy of MEK inhibition in combination with an mTOR inhibitor is in progress. Interestingly, both mTORC1 and mTORC2 inhibition induce a strong cytotoxic effect in MPNST cell lines [89]. More work is needed to determine if the combined inhibition of mTORC1 and mTORC2 is a more effective therapeutic strategy alone or in combination with MEK inhibition.

### 4.3. Wnt

Wingless/int (Wnt) signaling is an evolutionarily conserved signaling pathway known for its roles in embryonic development, cell fate specification, cell proliferation, and cell migration. Wnt canonical signaling is initiated by activating a cysteine-rich domain receptor named Frizzled (FZD) and its LRP5/6 co-receptors on the cell surface. In the absence of Wnt ligand, a protein complex including Axin, GSK3-β, adenomatosis polyposis coli (APC), and casein kinase 1 α (CK1α) sequesters β-catenin into the cell cytoplasm, targets it for phosphorylation and ubiquitination, and subsequently drives it to the proteasome for degradation. When bound to FZD, Wnt ligands induce Dishevelled (Dvl)/FZD intracellular interaction and subsequently enable Axin/APC/GSK3-β complex to bind and phosphorylate LRP5/6 [90]. As a result, β-catenin is released and can translocate from the cytoplasm into the nucleus, interact with T-cell factor and lymphoid enhancer factor (TCF/LEF), and induce transcription of multiple target genes including *MYC*, *CCND1*, and *BIRC5* [91,92,93,94]. MYC does not appear as an MPNST driver [11], but CCND1 and survivin (*BIRC5*) are at least associated with MPNST development [95,96,97].

As opposed to β-catenin N-terminal phosphorylation by GSK3-β, phosphorylation by CK2 in the armadillo repeat protein interaction domain prevents β-catenin degradation [98]. CK2 is overexpressed in MPNST cell lines, and the small molecule inhibitor CX-4945 slows tumor growth in an MPNST xenograft model [99]. However, some of its potency may be due to Wnt-independent signaling.

Other positive regulators of the Wnt pathway include the R-spondin (RSpo) family [100]. In fact, two homologous E3 ubiquitin ligases, Zinc and Ring Finger 3 (ZNRF3) and Ring finger protein 43 (RNF43), target FZD for ubiquitination, trigger internalization, and induce its degradation [94]. Furthermore, Dickkopf (DKK) proteins can downregulate LRP5/6 membrane expression by forming a ternary complex with Kremen receptors and triggering their internalization [100,101]. RSpo proteins inhibit ZNRF3 and RNF43 through LGR4/5 receptor activation [102], avoid DKKs-Kremen-LRPs association and thus, increase the membrane stability of FZD and LRP5/6 [100,101].

It has been reported that Wnt/β-catenin signaling is highly activated in many human cancers [94]. Aberrant activation could be a consequence of either Wnt ligand overexpression or mutations responsible for stabilization of β-catenin (β-catenin destruction complex inhibition or β-catenin activation). A systematic gene expression profiling of 89 components/regulators of the Wnt pathway in human MPNST indicates that 20 of them have altered expression compared to benign neurofibromas [103]. To demonstrate a cell-type specific expression pattern, the 20 gene set was examined in MPNST cell lines. Surprisingly, the Wnt pathway inhibitor DKK1 is about the most abundant and common gene expressed by the seven MPNST cell lines tested [103]. Interestingly, PRC2 regulates the Wnt pathway in breast cancer through DDK1 [104], but the PRC2-Wnt axis was not explicitly examined in MPNST.

Transcriptomic analysis of benign neurofibromas and MPNSTs from mouse models indicates that Wnt ligands have increased expression correlating with tumor progression, and components of the destruction complex of β-catenin are impaired most probably because APC and GSK3-β are downregulated [93]. Indeed, the staining of β-catenin seems to correlate with tumor aggressivity [93]. Expression of an active form of β-catenin impairs cell contact inhibition, and knocking down either the β-catenin gene CTNNB1 or TNKS (a known stabilizer of AXIN1) reduces cell viability anchorage-independent growth in vitro [93]. In addition, Watson et al. showed that RSpo2 is overexpressed in MPNST, and its down-regulation by shRNA leads to a decrease in Wnt signaling [93]. This result supports the hypothesis that multiple mechanisms in Schwann cells activate the Wnt pathway.

The Wnt pathway seems to play an important role in MPNST development, but its exact contribution remains undetermined. Surprisingly, the impact of genetically modulating the Wnt pathway in an MPNST mouse model has not been reported so far. Because recurrent mutations in the players of the Wnt pathway have not been reported thus far, it would be interesting to decipher how mutations in the genes leading to MPNST progression (i.e., *CDKN2A*, *EED*, *SUZ12*) activate the Wnt pathway, and whether these mechanisms can yield novel biomarkers of tumor stages. Because *SUZ12* is always disrupted in type 1 *NF1* microdeletion [4], it suggests that the *NF1* microdeletion modulates the Wnt pathway at least through SUZ12. Furthermore, the Wnt pathway can cross-talk with many other signaling pathways. PTEN loss-of-function promotes Akt activation, induces GSK3-β inhibition, and stabilizes β-catenin, ensuring MPNST development [17]. APC loss-of-function allows immortalized human Schwann cells to form a tumor in vivo, most probably because of its cross-talk with the Hippo pathway [105]. Thus, studying the Wnt pathway in the context of cross-talk with other MPNST-relevant pathways could enable the discovery of combination therapies.

## 5. Metastasis

We are now starting to unravel some important players in the metastatic process of MPNST cells. p21-activated kinases 1, 2, and 3 (PAK 1/2/3) are a protein family that influences Ras-driven cancer [106]. The activity of PAK1/2/3 is positively correlated with MPNST stage, and targeting PAK 1/2/3 decreases MPNST metastasis in mouse models [107]. Interestingly, PAK1/2/3 mainly modulates Akt and β-catenin activity with little influence on the MAPK pathway [107]. Galectin-1 is a small protein that helps in stabilizing activated Ras at the plasma membrane, which is essential to signal through the MAPK pathway [108]. Galectin-1 is highly expressed in MPNST, and its inhibition through small-molecule inhibitor decreases metastatic colonization in mouse models [109]. The RNA binding protein HuR was identified as a key player in the metastatic process of MPNST cells [110]. CAV1 mediates the suppressive function of MSI2 in epithelial-mesenchymal transition, migration, and invasion in vitro and metastasis in vivo [111]. Of note, one major caveat from these in vivo studies is the lack of a mouse model recapitulating the natural history of metastatic MPNST, which is highly desirable.

## 6. Conclusions

Following the discovery of the precursor lesion ANNUBP, a series of milestones were achieved, including guidelines for clinical care management, characterization of the genetic insult leading to ANNUBP, and the establishment of mouse models recapitulating human ANNUBP. More work is needed to characterize this precursor lesion at the cellular and molecular levels and determine its necessity for developing MPNST.

## Figures and Tables

**Figure 1 ijms-22-08620-f001:**
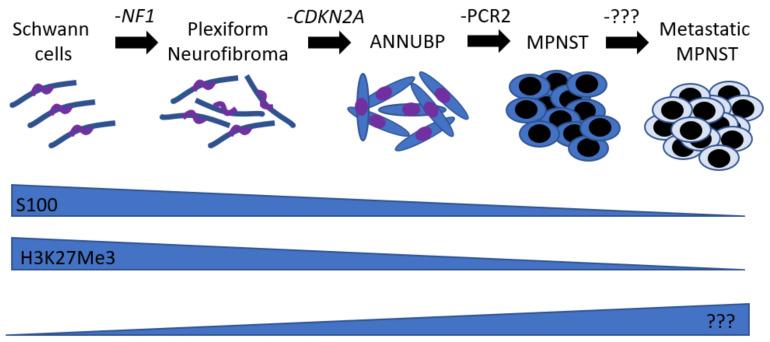
Neurofibroma to MPNST sequence. Schematic illustrating the genetic events driving the neurofibroma to MPNST sequence and the relative expression of markers associated with this transition.

**Figure 2 ijms-22-08620-f002:**
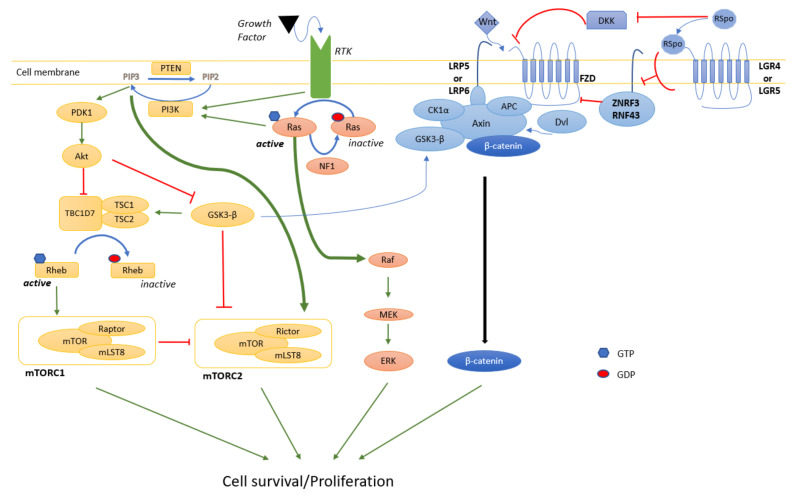
MPNST signaling pathways. Schematic illustrating the main pathways involved in the development/maintenance of MPNSTs (mTOR (yellow), MAPK (red), Wnt (blue)). Light blue arrows represent protein binding interaction. Dark blue arrows represent an enzymatic substrate to product conversion. Green arrows indicate activation, and red T means repression.

**Table 1 ijms-22-08620-t001:** Histological criteria to confirm the diagnosis of MPNST mouse models.

Mouse Model	Penetrance	Histological Criteria ^1^	Precursor Lesion?	Reference
cis*Nf1*^+/−^*p53*^+/−^	30%	- Schwann cell neoplasm (Presence of S100+ cells and micronerve) - Invasion of adjacent tissue (muscle)	Not observed	[26]
cis*Nf1*^+/−^*p53*^+/−^	N/A	- Schwann cell neoplasm (Presence of S100+ cells)	Not observed	[27]
Gfap-Cre *Pten* ^f/+^ LSL KrasG12D	100%	- Schwann cell neoplasm (Presence of S100+ cells) - Hypercellularity - Mitotic figures - Cytologic atypia (nuclei polymorphism) - Area of necrosis Ki67 positive	Neurofibroma	[16]
*Dhh-Cre* *Nf1* ^f/f^ *Pten* ^f/f^	N/A	- Schwann cell neoplasm (Presence of S100+ cells) Ki67 positive	N/A	[41]
*P_0_*-*GGF*b3	74%	- Schwann cell neoplasm (Presence of S100+ cells) - Hypercellularity Presence of mast cells	Neurofibroma	[38]
*Dhh-Cre**Nf1*^f/f^ *CNPase*-*hEGFR*	33%	- Schwann cell neoplasm (Presence of S100+ cells) Hypercellularity Mitotic figures - Cytologic atypia (nuclei polymorphism)	Neurofibroma	[40]
*Gfap-Cre**Nf1*^f/f^ OR *Gfap-Cre**Nf1*^f/^^−^ OR *Postn-Cre**Nf1*^f/f^ +pTOMOshp53	60–70%	- Schwann cell neoplasm (Presence of S100+ cells) Hypercellularity Mitotic figures - Cytologic atypia (nuclei polymorphism) Invasion of adjacent tissue Presence of mast cells (tryptase) Ki67 positive	Hyperplasia	[28]
*Nf1*^f/f^*Arf*^f/f^ OR *Nf1*^f/−^*Arf*^f/f^	N/A	- Schwann cell neoplasm (Presence of S100+ cells) - Presence of mast cells - Ki67 positive	Not observed	[44]
*Plp-*creERT2 *Nf1*^f/f^	10%	- Schwann cell neoplasm (Presence of S100+ micronerve) Hypercellularity Loss of H3K27Me3	Neurofibroma	[32]
*Postn-Cre* *Nf1*^f/f^*Arf*^f/f^ OR *Postn-Cre* *Nf1*^f/f^*Arf*^f/+^	100%	- Hypercellularity - Cytologic atypia (enlarged nuclei, hyperchromasia, pleomorphism) - Mitotic figures - Disruption of CD34 lattice	Neurofibroma and ANNUBP	[43]
*Dhh-Cre* *Nf1*^f/f^*Arf*^−^^/^^−^ OR *Dhh-Cre* *Nf1*^f/f^*Arf*^+/^^−^	40%	- Schwann cell neoplasm (Presence of S100+ cells and micronerve) - Cytologic atypia (nuclei polymorphism) Area of necrosis Invasion of adjacent tissue Presence of mast cells Ki67 positive	Neurofibroma and ANNUBP	[42]

^1^ In blue: Consensus criteria [45]. In red: Criterion not widely recognized or usually performed.

## Data Availability

Not applicable.

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
