# Peer review of "Toward Understanding the Mechanisms of Malignant Peripheral Nerve Sheath Tumor Development"

_ijms, 2021, doi:10.3390/ijms22168620_

Round 1
Reviewer 1 Report
The authors reviewed the literature on pathogenic causes underlying malignant peripheral nerve sheath tumors (MPNSTs) to shed light on the mechanisms involved in development of these tumors. After a description of the onset, the feature and MPNSTs development, the authors reported on the mouse models recapitulating MPNST development as well as the role of tumor environment and the involved signaling pathways. The paper is well written, nevertheless some criticisms are present because some important pathogenic mechanisms have been missed as indicated below.
MAJOR CRITICISMS
- It is well known that NF1 patients with NF1 microdeletion exhibit a more severe phenotype including the four- fold increased risk of MPNST. This topic is missing and should be included in the review, as well as the extensive supporting literature (Kehrer-Sawatzki et al., Am J Hum Genet 2004;75:410-423; Upadhyaya et al., Hum Genet 1998;102:591-597; Pasmant et al. Hum Mutat 2010;31:E1506-1518).
- NF1 type 1 microdeletion syndrome has implications on the gene dosage by genetic and epigenetic mechanisms causing hemizygosity of genes within the deletion such ATAD5, COPRS, UTP6 and RNF135 (Kehrer-Sawatzki et al Hum Genet. doi:10.1007/s00439-017-1766-y) and position effect on genes flanking the deletion (Ferrari et al., 2017, Hum Genet DOI 10.1007/s00439-017-1832-5), also contributing to increased tumor risk. These issues should be discussed
- The authors evidenced the role of SUZ12 to activate the WNT pathway, this is consistent with the fact that a copy of SUZ12 is always disrupted in type 1 NF1 microdeletion syndrome (Kehrer-Sawatzki et al., Am J Hum Genet 2004;75:410-423), associated to an increased risk of MPNST development; moreover, CDKN2A/B ANRIL locus deregulation has been also associated to increased risk of optic gliomas in NF1 patients (Zuccotti et al., GENES doi: 10.3390/genes10110892). These issues should be discussed.
MINOR REVISION
Natural Hystory: the sentence in lines 29-30 should be better explained with more details.
Author Response
The authors reviewed the literature on pathogenic causes underlying malignant peripheral nerve sheath tumors (MPNSTs) to shed light on the mechanisms involved in development of these tumors. After a description of the onset, the feature and MPNSTs development, the authors reported on the mouse models recapitulating MPNST development as well as the role of tumor environment and the involved signaling pathways. The paper is well written, nevertheless some criticisms are present because some important pathogenic mechanisms have been missed as indicated below.
MAJOR CRITICISMS
- It is well known that NF1 patients with NF1 microdeletion exhibit a more severe phenotype including the four- fold increased risk of MPNST. This topic is missing and should be included in the review, as well as the extensive supporting literature (Kehrer-Sawatzki et al., Am J Hum Genet 2004;75:410-423; Upadhyaya et al., Hum Genet 1998;102:591-597; Pasmant et al. Hum Mutat 2010;31:E1506-1518).
We thank the reviewer for pointing our attention to this topic. On page 2. Line 45, we add the following paragraph that included the suggested citations:
`` Beside NF1 microdeletion, a large deletion of 1.4 Mb encompassing the NF1 gene [5-7] occurring in up to 10% of NF1 patients [8], there is no specific highly recurrent chromosomal abnormality in MPNST cancer cells. Interestingly, patients with NF1 microdeletion have a 16 – 26% risk of developing MPNST [9] (compared to a 8-13% risk of developing MPNST for NF1 patients regardless of the specific NF1 mutation [10]. There is at least 14 protein coding genes and 4 miRNAs encoded in the microdeletion region. This led to the hypothesis that the increased MPNST risk may be due to the loss of additional tumor suppressor genes. A decade of effort culminates into the demonstration that SUZ12 loss, one of the 14 protein coding genes, promotes MPNST progression in mouse model [11]. In fact, mutations in components of the Polycomb repressive complex 2 (PRC2) complex (i.e. EED and SUZ12) are associated with MPNST [12, 13]. The majority of MPNSTs also harbor somatic mutations in the cell cycle regulator CDKN2A, regardless of their etiology [4]. ``.
- NF1 type 1 microdeletion syndrome has implications on the gene dosage by genetic and epigenetic mechanisms causing hemizygosity of genes within the deletion such ATAD5, COPRS, UTP6 and RNF135(Kehrer-Sawatzki et al Hum Genet. doi:10.1007/s00439-017-1766-y) and position effect on genes flanking the deletion (Ferrari et al., 2017, Hum Genet DOI 10.1007/s00439-017-1832-5), also contributing to increased tumor risk. These issues should be discussed
As suggested by the reviewer, we now included (page 3, line 79) a discussion of the contribution of other genes that are part of the NF1 microdeletion and their impact on the risk of developing MPNST
`` Kehrer-Sawatzki et al. [8] pointed out that there are at least three other genes with tumor suppressor activity within the microdeletion region: miR193A, miR365B, and the ATPase family AAA domain-containing protein 5 (ATAD5). Although there is a wide body of evidence of their tumor suppressor activity in other cancer types, there is currently no report of the direct contribution of any of these three genes in the context of MPNST development. Importantly, there could be a significant position effect on genes flanking the microdeletion that may ultimately promote MPNST progression. For example, a NF1 microdeletion could produces a chimeric gene that results in surrounding genes being overexpressed [19].``
- The authors evidenced the role of SUZ12 to activate the WNT pathway, this is consistent with the fact that a copy of SUZ12 is always disrupted in type 1 NF1 microdeletion syndrome (Kehrer-Sawatzki et al., Am J Hum Genet 2004;75:410-423), associated to an increased risk of MPNST development; moreover, CDKN2A/B ANRIL locus deregulation has been also associated to increased risk of optic gliomas in NF1 patients (Zuccotti et al., GENES doi: 10.3390/genes10110892). These issues should be discussed.
Thank you for the suggestion. On page 14, line 394, we add the following sentence to discuss these issues. Since the review focus on MPNST, we omitted the suggested reference related to optic gliomas.
``Since SUZ12 is always disrupted in type 1 NF1 microdeletion [4], it suggests that the NF1 microdeletion modulates the wnt pathway at least through SUZ12.``
MINOR COMMENTS
Natural History: the sentence in lines 29-30 should be better explained with more details
We clarify the sentence in lines 29-30
``Since MPNSTs are tumors affecting sensitive nerves, it is not rare for these patients to experience numbness in their extremities or having difficulty moving their limbs (paraparesis).``
Overall, we are grateful for the critical review of the manuscript and we believe we now have an acceptable manuscript version for publication.
Reviewer 2 Report
The review is well written and summarizes key findings and challenges in MPNST research. A few minor suggestions:
1) The summary of the available mouse models of MPNST is very helpful. The cell of origin of MPNST remains a very interesting topic. Can the authors provide more insights into the cell of origin based on the various Cre strains used to model MPNST? Also, insights into commonly used MPNST cell lines?
2) Some discussion about the role of the Nf1 heterozygous stromal cells in MPNST pathogenesis would be helpful.
3) There are some unwanted red lines in Figure 1.
Author Response
Reviewer 2 :
The review is well written and summarizes key findings and challenges in MPNST research. A few minor suggestions:
1) The summary of the available mouse models of MPNST is very helpful. The cell of origin of MPNST remains a very interesting topic. Can the authors provide more insights into the cell of origin based on the various Cre strains used to model MPNST? Also, insights into commonly used MPNST cell lines?
We thank the reviewer for the interest in our manuscript. The cell of origin of MPNST is indeed a very interesting research subject. Even if various Cre (Gfap-cre, plp-creERT2, Dhh-cre, postn-cre) were used, their direct comparison would be difficult. Therefore, we did not further discuss the cell of origin of MPNST.
2) Some discussion about the role of the Nf1 heterozygous stromal cells in MPNST pathogenesis would be helpful.
Good point. On page 10, line 266, we add a discussion on Nf1 +/- stromal cells.
``As discussed by Brosseau and collaborators [45, 46], it is known that the Nf1 +/- microenvironment contributes to neurofibroma development, but the underlying molecular and cellular mechanisms are elusive. Even less is known about the contribution of the Nf1 +/- microenvironment on MPNST development. Initial studies indicate that an Nf1 +/- microenvironment promotes MPNST development [44, 47]. A CD45+ population of immune cells may possess tumor-promoting capacities [44]. However, a recent study using two orthogonal mouse models indicates that the Nf1 +/- microenvironment impairs malignant progression [32]. Since, Nf1 +/- CD8+ T cells trigger an enhanced immune reaction and that Nf1 +/- natural killer cells have a protective effect compared to their wild type counterpart, it raises the intriguing possibility that Nf1 +/- T cells may explain the impairing phenomenon observed by Brosseau et al. [32]. More works are needed to dissect the anti and pro-tumorigenic properties of the Nf1 +/- microenvironment to MPNST development. ``
3) There are some unwanted red lines in Figure 1.
Good catch. We removed the red lines.